# OpenReview forum: "Convergence Towards Stable Intrinsic Self-correction of Large Language Models"
_ICLR.cc/2025/Conference — Submitted to ICLR 2025_

### Official Review · Reviewer_AkNZ · 2024-10-28

**Soundness:** 2
**Presentation:** 2
**Contribution:** 2
**Rating:** 5
**Confidence:** 2

**Summary:**

The author's theoretically analyze self-correction and establish a connection to model uncertainty and calibration error.  They empirically demonstrate that model uncertainty and calibration error converge over multiple rounds of self-correction on several QA and generation tasks.

**Strengths:**

1. The paper provides theoretical grounding for self-correction
1. The empirical results are tested over a two tasks (QA & generation) and across several domains each.

**Weaknesses:**

1. The text based tasks are only evaluated with Mistral-7B-v0.1 but having results for other model families or sizes could improve the robustness of the empirical results.
2. There is a gap between the argument of the connection between uncertainty and calibration error, and the empirical results.  Specifically, the paper could address the fact that the QA converge in calibration error and performance after one round of iteration while the uncertainty only drops by a small amount after that same round.

**Questions:**

In Figure 3 we see that 2 of the QA tasks converge in performance after a single round of self-correction.  This is confirmed in Figure 4b where the calibration error also converges after a single round of self-correction.  However, Figure 4a shows that uncertainty only drops by a small amount after a single round, and only converges after +10 rounds of correction.  This calls into question the crux of the paper that mechanism of self-correction performance improvements are driven by lowered uncertainty.  After a single round of self-correction the model can "re-read" the question (see *Re-Reading Improves Reasoning in Large Language Models*, Xu et al. 2024) which may account for the single-shot convergence of performance.

The paper could be improved by addressing this empirical inconsistency that error converges after a single iteration round but uncertainty doesn't converge until many more rounds.

---

> ### Author Response · Authors · 2024-11-22
>
> Dear reviewer, we really appreciate your comments and great suggestions.
>
> **Weakness1:** The text based tasks are only evaluated with Mistral-7B-v0.1 but having results for other model families or sizes could improve the robustness of the empirical results.
>
> **Response:** (1) From a technical point of view, the 7B Mistral model is the only model which has two key traits required for our study: (1) 7B Mistral has demonstrated strong performance in following-type instructions, and (2) researchers have access to a version without safety alignment. LLMs with safety alignment may impair a model's ability for moral self-correction [1], thus limiting researchers’ ability to evaluate its intrinsic capabilities. Again, to the best of our knowledge, 7B Mistral is the only model that meets these specific requirements. We have also noted this in our limitations section.
> (2) From a research point of view, we conducted a comprehensive study of intrinsic moral self-correction and revealed the convergence nature of intrinsic moral self-correction using the 7B model. Currently, there are many influential papers exploring the internal mechanisms of LLMs and these papers use only one model for experiments, just as we did. Two representative works are [2,3]. Similar to these works, we attend to the general patterns of the behaviors of LLMs, and are not restricted to any specific architecture. Therefore, the generalizability of our conclusion holds.
>
> But we do believe larger LLMs would enjoy better convergence rate and final performance.
>
> [1] Ganguli, Deep, et al. "The capacity for moral self-correction in large language models." arXiv preprint arXiv:2302.07459 (2023).
>
> [2] Lee, Andrew, et al. "A Mechanistic Understanding of Alignment Algorithms: A Case Study on DPO and Toxicity." Forty-first International Conference on Machine Learning.
>
> [3] Geva, Mor, et al. "Transformer Feed-Forward Layers Are Key-Value Memories." Proceedings of the 2021 Conference on Empirical Methods in Natural Language Processing. 2021.
>
> **Weakness2:** There is a gap between the argument of the connection between uncertainty and calibration error, and the empirical results. Specifically, the paper could address the fact that the QA converge in calibration error and performance after one round of iteration while the uncertainty only drops by a small amount after that same round.
>
> **Response:** In this draft, we aim to establish connections among self-correction instructions, activated concepts, uncertainty, calibration error, and convergence. The causal relationship between uncertainty and calibration error is validated through previous studies[1,2,3] as we shown in the section 4. As illustrated in Figure 2, our focus is on bridging the gap between activated concepts and convergence using both empirical and theoretical evidence. We provide empirical support for the relationships between each pair of variables (such as section 6.1) and offer theoretical evidence to link instructions to convergence, we carefully convey claims to avoid any misleading. Given the extensive and long chain of the logical reasoning (Figure 2), from the instruction for self-correction to the endpoint of convergence, **we avoid reiterating conclusions from prior studies to ensure the draft remains focused and information-rich**.
>
> [1] Arendt, Paul D., Daniel W. Apley, and Wei Chen. "Quantification of model uncertainty: Calibration, model discrepancy, and identifiability." (2012): 100908.
>
> [2] Wang, Deng-Bao, Lei Feng, and Min-Ling Zhang. "Rethinking calibration of deep neural networks: Do not be afraid of overconfidence." Advances in Neural Information Processing Systems 34 (2021): 11809-11820.
>
> [3] Ao, Shuang, Stefan Rueger, and Advaith Siddharthan. "Two sides of miscalibration: identifying over and under-confidence prediction for network calibration." Uncertainty in Artificial Intelligence. PMLR, 2023.

---

> > ### Author Response · Authors · 2024-11-22
> >
> > **Question1:** In Figure 3 we see that 2 of the QA tasks converge in performance after a single round of self-correction. This is confirmed in Figure 4b where the calibration error also converges after a single round of self-correction. However, Figure 4a shows that uncertainty only drops by a small amount after a single round, and only converges after +10 rounds of correction. This calls into question the crux of the paper that mechanism of self-correction performance improvements are driven by lowered uncertainty. After a single round of self-correction the model can "re-read" the question (see Re-Reading Improves Reasoning in Large Language Models, Xu et al. 2024) which may account for the single-shot convergence of performance. The paper could be improved by addressing this empirical inconsistency that error converges after a single iteration round but uncertainty doesn't converge until many more rounds.
> >
> >
> > **Response:** Thank you for your excellent question and for suggesting additional reading. Our response is twofold:
> >
> > (i) **Inconsistencies Induced by Metrics**. The primary source of inconsistency lies in the noise introduced by different metrics, particularly semantic uncertainty, which induces more noise during measurement. We appreciate your careful observation that convergence does not occur at the same round.
> >
> > The main reason is that the uncertainty measure (semantic uncertainty we adopt) requires a high temperature to control the randomness of responses, ensuring varied responses for uncertainty measurement. However, in most other experiments, we set the temperature to 0 to ensure reproducibility, which leads to faster convergence. We will add an explanation regarding this discrepancy in our revision.
> >
> > (ii) **Discussion on the suggested Related Work**
> > Thank you for the suggestion to include the proposed paper in the related work section. The strategy outlined in the paper shares a similar spirit with our approach. While our method repeats the instruction, the proposed method repeats the inputs. We hypothesize that both methods aim to activate positive concepts while reducing uncertainty. Additionally, the proposed paper reinforces the broader implications of our LLM mechanism analysis. We will include a discussion on this in the revised version.

---

> > > ### Author Response · Authors · 2024-11-26
> > >
> > > Dear Reviewer, as the discussion period is nearing its end, we would be happy to address any remaining questions you may have.

---

### Official Review · Reviewer_k7kS · 2024-10-29

**Soundness:** 3
**Presentation:** 4
**Contribution:** 3
**Rating:** 8
**Confidence:** 4

**Summary:**

This paper investigates the mechanisms of convergence in 'self-correction' in LLMs. By monitoring concepts, uncertainty, and toxicity of the responses, the automatic feedback will prompt the LLM in contexts to improve in a specific dimension. With multiple-round interactions, the LLM can converge across these three dimensions in its response. With this design, the authors analyzed how the model's performance and those metrics change over interactions in six different tasks. The results show that all three metrics converged and were lower than the baseline, revealing 'self-improvement' in model responses. Finally, the authors show theoretical proof of the connection between model uncertainty and concept representation, and how such iterative correction processes could lead to convergence. Overall, this paper provides a deep insight into the mechanisms of 'self-correction' in LLMs.

**Strengths:**

- The paper chooses six different tasks to test the model performance, which varies across QA tasks and text generations, as well as multi-modal visual understanding tasks, showing a wide distribution of the scenarios.

- The paper dives into the mechanistic explanations of how and why 'self-correction' can converge in iterations. The work integrates both empirical and theoretical foundations to explain the phenomenon.

- The writing and visualization of the paper is satisfying.

**Weaknesses:**

- Only one model is used. I was wondering how different models (e.g., one mode family with different sizes; or different model family with the same size) could perform differently. This may help us understand deeper why the model could improve and what may influence the speed of convergence as well as the optimal point.

- Model uncertainty measurement is good since it does not rely on external resources and can gain complete accessibility from the model itself. The latent concept is also flexible and does not rely on external benchmarks. However, EC relies on the task ground truth and label data. This means that, in unseen tasks, the model may not be able to get feedback from EC, which may affect the generalizability of the task. Therefore, I wonder if there is any experimental setting or analysis to show that without EC feedback, how can the pipeline solely rely on model uncertainty and latent concept can reach the performance; or how much does the three dimensions of feedback contribute to a convergent process.

- (Minor) There are some questions remaining unclear regarding the convergence. For example, in the process, how the 'gradient' like concepts are represented in the model(which means how does the model know which direction to correct), from contexts to model computations. This may help us understand better how the model perceives (evaluates) these metrics. Another question is, what determines the convergence optimal point? (model size? or complexity of the task? human-instruction fine-tuning?). Some exploratory analysis on these aspects would much improve the depth of this paper. (if these questions are somehow tackled, I believe this paper should be at least 8 or even 10 score level).

**Questions:**

- I wonder how general this probing vector could be. I did not put this into weakness because it looks great in the paper. However, I still wonder only with this probing vector, how can it guide the performance in benchmark-free scenarios.

---

> ### Author Response · Authors · 2024-11-22
>
> Dear reviewer, we really appreciate your helpful comments and insightful suggestions for our draft.
>
> **Weakness1.** Only one model is used. I was wondering how different models (e.g., one mode family with different sizes; or different model family with the same size) could perform differently. This may help us understand deeper why the model could improve and what may influence the speed of convergence as well as the optimal point.
>
> **Response:** (1) From a technical point of view, the 7B Mistral model is the only model which has two key traits required for our study: (1) 7B Mistral has demonstrated strong performance in following-type instructions, and (2) researchers have access to a version without safety alignment. LLMs with safety alignment may impair a model's ability for moral self-correction [1], thus limiting researchers’ ability to evaluate its intrinsic capabilities. Again, to the best of our knowledge, 7B Mistral is the only model that meets these specific requirements. We have also noted this in our limitations section.
> (2) From a research point of view, we conducted a comprehensive study of intrinsic moral self-correction and revealed the convergence nature of intrinsic moral self-correction using the 7B model. Currently, there are many influential papers exploring the internal mechanisms of LLMs and these papers use only one model for experiments, just as we did. Two representative works are [2,3]. Similar to these works, we attend to the general patterns of the behaviors of LLMs, and are not restricted to any specific architecture. Therefore, the generalizability of our conclusion holds.
> But we do believe larger LLMs would enjoy better convergence rate and final performance.
>
> [1] Ganguli, Deep, et al. "The capacity for moral self-correction in large language models." arXiv preprint arXiv:2302.07459 (2023).
>
> [2] Lee, Andrew, et al. "A Mechanistic Understanding of Alignment Algorithms: A Case Study on DPO and Toxicity." Forty-first International Conference on Machine Learning.
>
> [3] Geva, Mor, et al. "Transformer Feed-Forward Layers Are Key-Value Memories." Proceedings of the 2021 Conference on Empirical Methods in Natural Language Processing. 2021.
>
> **Weakness2:** Model uncertainty measurement is good since it does not rely on external resources and can gain complete accessibility from the model itself. The latent concept is also flexible and does not rely on external benchmarks. However, EC relies on the task ground truth and label data. This means that, in unseen tasks, the model may not be able to get feedback from EC, which may affect the generalizability of the task. Therefore, I wonder if there is any experimental setting or analysis to show that without EC feedback, how can the pipeline solely rely on model uncertainty and latent concept can reach the performance; or how much does the three dimensions of feedback contribute to a convergent process.
>
> **Response:** Thank you for your question. Nonetheless, we are still unsure about what "EC" you are referring to. We hypothesize that EC might mean ECE (Expected Calibration Error) or Calibration Error, which typically requires task ground truth. In our approach, we use simple prompts such as "Do not be biased" as the input. Latent concepts, uncertainty, and ECE are employed as metrics to evaluate the quality of convergence, but **they are not used as feedback to guide the convergence process**. The model does not rely on any task ground truth. Instead, the simple self-correction prompt results in increased positive concepts, reduced model uncertainty, and decreased calibration error. Calibration Error is used to assess whether the reduced uncertainty improves downstream performance.

---

> > ### Author Response · Authors · 2024-11-22
> >
> > **Weakness3.1:** (Minor) There are some questions remaining unclear regarding the convergence. For example, in the process, how the 'gradient' like concepts are represented in the model(which means how does the model know which direction to correct), from contexts to model computations. This may help us understand better how the model perceives (evaluates) these metrics.
> >
> > **Response**: Thanks for your interesting question regarding a more fine-granularity analysis of how different model components contribute to activating positive concepts.vIn [1], a simplified setup is adopted to demonstrate that self-correction samples can be used to generate responses with higher rewards aligned with positive concepts. The forward process in each layer can be interpreted as a gradient step towards optimizing a Bradley-Terry model.vWe see this as a promising direction for future work. However, it is important to note that the mentioned analyses often rely on numerous simplifications and experiments on synthetic data, whereas most observations in our paper focus on practical applications across multiple real-world scenarios. vWe would like to discuss about this direction as the future work. Notably, there is a significant gap between such analysis with our paper. Those analyses require many simplifications with experiments on synthetic datas. Our paper is more practical with multiple real-world scenarios. In our revision, We add one section to show how our model can help effectively select samples for finetuning to improve the self-correction capability.
> >
> > [1] Wang, Yifei, et al. "A Theoretical Understanding of Self-Correction through In-context Alignment." NeurIPS (2024)
> >
> > **Weakness3.2:** Another question is, what determines the convergence optimal point? (model size? or complexity of the task? human-instruction fine-tuning?). Some exploratory analysis on these aspects would much improve the depth of this paper. (if these questions are somehow tackled, I believe this paper should be at least 8 or even 10 score level).
> >
> > **Response:**
> >
> > We appreciate this great and fundamental question about optimal point of convergence. We guess you mean the source of self-correction capability, and how could we accelerate convergence.
> >
> > In terms of the inference process, self-correction is straightforward and simple since it takes input and make a conditional likelihood calculation to the desired output. However, this straightforward viewpoint is uninformative for understanding self-correction. Instead, from a mechanistic view, the convergence is driven by the activated concepts, which can be considered as (1) the function from the learned function classes during pretraining or as (2) the manifold underlying the activations (hidden states) across layers. Therefore even though the input self-correction instruction activats concepts, but this concept depends on pre-training data which is unknown to us. However, we can definitely enhance self-correction (through fine-tuning) by re-aligning LLMs with new data ditributions. Our hypothesis is supported by several recent studies that were published after our work.
> >
> > Recently, [1] concludes that self-correction is not an innate capability of LLMs, e.g. In-context Learning, as they can not detect the difference among their outputs, therefore it can be enhanced through additional fine-tuning. [2,3] shows two RL-based solution for enhancing intrinsic self-correction through additiona fine-tuning. However, their solutions still pursue a superficial but effective way by showing self-correction demonstrations to LLMs but do not present any insights about how this capability emerges during pre-training. The primary reason for this delimma is **self-correction is a reflection of the instruction-following capability** of LLMs, the instruction-following capability is a fundamental capability but is not well-explored at all. We also discuss those points in section 7&8.
> >
> > Due to the complexity of LLMs, purely theoretical and empirical studies often overlook concerns associated with their respective counterparts. Our draft bridges this gap and stands among the few to explore the mechanism of self-correction in depth. While further efforts are needed to uncover the nature of self-correction, our work makes a critical contribution by demonstrating how instructions lead to a significant characteristic of intrinsic self-correction: convergence.
> >
> > [1] Qi, Zimo, et al. "Is Moral Self-correction An Innate Capability of Large Language Models? A Mechanistic Analysis to Self-correction." arXiv preprint arXiv:2410.20513 (2024).
> >
> > [2] Kumar, Aviral, et al. "Training language models to self-correct via reinforcement learning." arXiv preprint arXiv:2409.12917 (2024).
> >
> > [3] Qu, Yuxiao, et al. "Recursive introspection: Teaching language model agents how to self-improve." arXiv preprint arXiv:2407.18219 (2024).

---

> > > ### Author Response · Authors · 2024-11-22
> > >
> > > **Question1:** I wonder how general this probing vector could be. I did not put this into weakness because it looks great in the paper. However, I still wonder only with this probing vector, how can it guide the performance in benchmark-free scenarios.
> > >
> > > **Response:** The generaliability of this probing vector is good but we still need the coverage of the corpus to extract the probing vector for sure. The reason that we are positive to this probing vector in our cases is we focus on semantics-intensive tasks this is the advantages of LLMs. However, we would doubt the effectiveness of the probing vector if we target pragmatics-level tasks such as framing analysis, ethics, etc.
> > >
> > > In the wild, users' queries are very short though they are harmful. If the corpus for getting the probing vector can cover those cases, the probing vector can be used for a real product as we present in the section 6.1

---

> > ### Comment · Reviewer_k7kS · 2024-11-25
> >
> > Dear authors,
> >
> > Thanks for your clarification. I misunderstood the feedback circle and thought those variables (e.g., ECE, uncertainty or so) are provided into the feedback loop. Given they are only used for evaluation, I would say this is an interesting approach with a strong generalization method with only prompts like 'do not be biased'. Since I originally think it a main weakness, I would like to raise the ratings and confidence.

---

> > > ### Author Response · Authors · 2024-11-26
> > >
> > > Dear reviewer,
> > >
> > > We really appreciate your feedback. Thanks so much of your comments in helping us to have a better draft. We are sure to cover our discussion in the future version.
> > >
> > > Best,
> > > All authors

---

### Official Review · Reviewer_dhoA · 2024-10-30

**Soundness:** 3
**Presentation:** 2
**Contribution:** 2
**Rating:** 3
**Confidence:** 4

**Summary:**

This paper presents an analysis of the self-correction approach to improving LLM responses. The authors find that model uncertainty decreases and a relevant concept seems to become more activated through the self-correction process.

**Strengths:**

The paper is focused on understanding a phenomenon and uses multiple tools to do so. This potentially gives us deeper insight into why this technique is effective.

**Weaknesses:**

1. The methods for measuring uncertainty were unclear. The various methods were stated with references but should be defined in the text for them to be interpretable.

2. No statistical significance testing was performed to support any of the claims about changes in uncertainty or activation of the latent concept. No error bars appear on plots. For this reason the latent concept analysis in particular seems relatively superficial.

3. The theoretical analysis makes strong independence assumptions: that question, instruction, and output are independent. With these assumptions it is not a surprise that convergence is geometric. I would like to see a more strongly motivated model.

4. The authors note that they did joy study other domains as self-correction is less effective there. That seems like an important omission — if the current approach helps us understand the efficacy of self-correction it should also explain the cases where it doesn’t work.

**Questions:**

I had no further questions.

---

> ### Author Response · Authors · 2024-11-22
>
> Dear reviewer, we really appreciate your comments and great feedback.
>
> **Weakness1**: The methods for measuring uncertainty were unclear. The various methods were stated with references but should be defined in the text for them to be interpretable.
>
> **Answer:** To make it more clear about **uncertainty measurement**, we use (1) the semantic uncertainty[1] for language generation task (e.g., text detoxicification); (2) prediction logits[2], approximated by log-likelihood of each token sequence of answer choice, for multi-choice QA tasks (e.g., social bias mitigation) by considering multi-choice QA tasks as classification tasks.
>
> To estimate the **calibration error**, we take the rank-calibration[3] for language generation tasks and the ECE error[4] for multi-choice QA tasks.
>
> **Weakness2**: No statistical significance testing was performed to support any of the claims about changes in uncertainty or activation of the latent concept. No error bars appear on plots. For this reason the latent concept analysis in particular seems relatively superficial.
>
> **Answer:** Thanks for this suggestion, we will add the error bar in our future version. However, the method we use to measure latent concepts has been validated in other accepted papers[5,6]. Our primary focus is on the trend across experimental settings, and we believe the current mean similarity results are sufficiently expressive.
>
> **Weakness3**: The theoretical analysis makes strong independence assumptions: that question, instruction, and output are independent. With these assumptions it is not a surprise that convergence is geometric. I would like to see a more strongly motivated model.
>
> **Answer:** The independence assumption is because
> (1) currently the research community still does not have a solid theoretical tool to interpret LLMs which remain a black-box for us. In-context learning is an example, some studies consider it as a mesa optimization but others believe it implements a meta-learning process. [7] proposes a theoretical analysis to the self-correction by assuming self-correction process implements a Bradley-Terry model through gradient descent. The authors have an assumption about the equivalence between LLMs' (forward) inference and gradient descent. In our case, the assumption of independence is the most optimal and effective one we can make right now.
> (2) we do not pursue to have an accurate mathematical modeling for how self-correction works, instead, based on our empirical observations, we aim at having a theoretical modeling to link the activated concept and the convergence of intrinsic self-correction. Therefore, we can ensure the relationship between activated concept and the convergence of self-correction is acceptable both empirically and theoretically. Our way for interpreting the convergence of self-correction is based on our empirical observations.
>
> **Weakness4**: The authors note that they did joy study other domains as self-correction is less effective there. That seems like an important omission — if the current approach helps us understand the efficacy of self-correction it should also explain the cases where it doesn’t work.
>
> **Answer:** Yes, reasoning has been a challenge for self-correction as we highlight in the section 7. We focus on semantics-intensive tasks in this draft is because (1) why self-correction can work remains unsolved even for semantics-intensive tasks (2) reasoning requires more beyond the powerful semantic modeling capability of LLMs.
>
> In terms of the inference process, self-correction is straightforward and simple since it takes input and make a conditional likelihood calculation to the desired output. However, this straightforward viewpoint is uninformative for understanding self-correction. Instead, from a mechanistic view, the convergence is driven by the activated concepts, which can be considered as (1) the function from the learned function classes during pretraining or as (2) the manifold underlying the activations (hidden states) across layers. Therefore even though the input self-correction instruction activates concepts, but this concept depends on pre-training data which is unknown to us. For the considered tasks in our draft, we leverage the probing vector to estimate the function or manifold, underlying the representation, by taking the advantage of semantic modeling of LLMs. For reasoning tasks and pragmatics-level tasks, we need more efforts for interpreting the effectiveness of self-corrections since those tasks requires more beyond only semantics.

---

> > ### Author Response · Authors · 2024-11-22
> >
> > [1] Kuhn, Lorenz, Yarin Gal, and Sebastian Farquhar. "Semantic Uncertainty: Linguistic Invariances for Uncertainty Estimation in Natural Language Generation." The Eleventh International Conference on Learning Representations.
> >
> > [2] Wu, Huiyu, and Diego Klabjan. "Logit-based uncertainty measure in classification." 2021 IEEE International Conference on Big Data (Big Data). IEEE, 2021.
> >
> > [3] Huang, Xinmeng, et al. "Uncertainty in language models: Assessment through rank-calibration." arXiv preprint arXiv:2404.03163 (2024).
> >
> > [4] Guo, Chuan, et al. "On calibration of modern neural networks." International conference on machine learning. PMLR, 2017.
> >
> > [5] Lee, Andrew, et al. "A Mechanistic Understanding of Alignment Algorithms: A Case Study on DPO and Toxicity." Forty-first International Conference on Machine Learning.
> >
> > [6] Liu, Guangliang, et al. "Intrinsic Self-correction for Enhanced Morality: An Analysis of Internal Mechanisms and the Superficial Hypothesis." Proceedings of the 2024 Conference on Empirical Methods in Natural Language Processing. 2024.
> >
> > [7] Wang, Yifei, et al. "A Theoretical Understanding of Self-Correction through In-context Alignment." arXiv preprint arXiv:2405.18634 (2024).

---

> > ### Comment · Reviewer_dhoA · 2024-11-25
> >
> > Thank you for the thoughtful response. Unfortunately I think the lack of statistical significance testing is still very relevant even if the goal is to compare trends -- without any form of quantification of the error in results it is impossible to know if the observed trends are meaningful. I appreciate the authors' responses but I think the weaknesses I identified still stand.

---

> > > ### Author Response · Authors · 2024-11-26
> > >
> > > Dear reviewer, thanks for your feedback.  For the error bar, as we promised please refer to the Figure 5 in the revised version. Regarding the significance test, **we kindly ask which aspects of Figure 5 you are specifically interested in regarding significance testing**.
> > >
> > > Please note that the benchmarks used in our paper are not small. For example, the RealToxicity benchmark consists of 99,442 samples, while the BBQ benchmark includes datasets on various social bias dimensions with sample sizes ranging from approximately 432 to 1,500. Additionally, we adopt the common deterministic setting in LLMs' inference with a greedy decoding.

---

### Official Review · Reviewer_43RE · 2024-11-01

**Soundness:** 4
**Presentation:** 2
**Contribution:** 3
**Rating:** 8
**Confidence:** 3

**Summary:**

The paper analyzes in detail the phenomenon of intrinsic self-correction, whereby an LLM is iteratively prompted using a pre-selected prompting strategy (in a multi-turn conversation without requiring human intervention) to improve its responses on a specific metric (e.g. toxicity, safety). The paper examines the convergence of this process in a multi-turn setting both by evaluating the original metric on the response (e.g. toxicity) and by looking at key metrics associated with the model’s internal representation (uncertainty and concept, which is a quantifiable way of probing the model’s internal “direction”) and show that they improve and converge during the intrinsic self correction. The paper studies the process convergence for several different tasks, which is crucial for obtaining quality guarantees in real-world use cases.

The paper shows that this phenomenon arises along with a reduction in the model’s uncertainty as part of the intrinsic self-correction trajectory, and further that this generates calibrated responses (in the sense that the model’s uncertainty is aligned with the correctness of the answer), with the initial responses tend to be under-confident. The paper formalizes the procedure’s steps as classification (e.g. yes/no) questions and uses normalized log-likelihoods to assess model’s certainty.
Additionally, the paper measures the latent concept’s development throughout the self-correction process. The latent concept is modeled as a binary positive/negative set (e.g. in biasing or discrimination - negative means exhibiting stereotypes or discrimination, and positive means fairness). This is measured using a linear probing vector to find the “direction” representing the concept in activation space. It is shown that along the self-correction trajectory it both converges and does not revert from positive to negative.
Finally, the paper performs an empirical and theoretical analysis to establish the causal relationship between the concept’s development and uncertainty reduction throughout the process, leading to convergence

**Strengths:**

The authors present a unified framework for assessing the convergence of intrinsic self-correction through multi-turn automatic prompting across multiple tasks. By carefully measuring both the model’s certainty throughout the process and the concept that the process is meant to improve (using a succinctly-described bayesian inference framework, and measured using a trained linear probe) they are able to show that both reach convergence across several tasks and that they do so in a monotonic way.
Further ablation studies demonstrate that, indeed, it was the specific prompting strategy that drove this, as injecting “opposite direction” prompts show the opposite behavior in the concept probe.
The technique is applied both to LLM and VLMs (i.e. for visual tasks). Both are shown to both be improved by the process and show the same trends in internal representation analysis.
Finally, the simulation-based analysis which links together uncertainty and concept is very interesting, and presents strong evidence for causality, later supported by theoretical analysis.

**Weaknesses:**

The paper could benefit from rewriting to improve clarity.
Specifically, consider rewriting the mathematical part of section 6.2, since it is difficult to follow and raises some questions specifically about the notation and derivation leading to eq. 3, discussed below,

**Questions:**

Mathematical analysis:
* In the derivation of equation 3, is it assumed that c_y and c_i are identical for all the steps? (If not, please make sure the notation clarifies that. )
* Is cx ≪ ci used anywhere?
* In Equation 1, is it assumed implicitly that there is only one concept (and the integral is thus converted into a sum over the negative and positive concept? Is this a trivial observation?

Very small comment
 “We leverage u2 − u1 as the change of concept and the label is set as 1 if u2 − u1 is no larger than 0, otherwise the label should be 0.” this should probably be a label for uncertainty, not concept, right?

---

> ### Author Response · Authors · 2024-11-22
>
> Dear reviewer, we really appreciate your comments.
>
> **Weaknesses:**
> The paper could benefit from rewriting to improve clarity. Specifically, consider rewriting the mathematical part of section 6.2, since it is difficult to follow and raises some questions specifically about the notation and derivation leading to eq. 3, discussed below,
>
> **Answer:** Thanks for your suggestion, we improve the section 6.2 in the revised version.
>
> **Questions on Mathematical analysis**:
>
> **Question1:** In the derivation of equation 3, is it assumed that c_y and c_i are identical for all the steps? (If not, please make sure the notation clarifies that. )
>
> **Answer:** According to our independence assumptions, c_i is identical but c_y is not. We have updated this in the revised version.
>
> **Question2:** Is cx ≪ ci used anywhere?
>
> **Answer:** Thanks for this question, we remove this sentence.
>
> **Question3:** In Equation 1, is it assumed implicitly that there is only one concept (and the integral is thus converted into a sum over the negative and positive concept? Is this a trivial observation?
>
> **Answer:** To be honest, the activated concepts should not only be of one concept, i.e., toxicity, but can be anything underlying the input query. Considering that LLMs have become an interface that can handle all information-seeking inqueries, it would be great to have analysis to a set of concepts. Though it is reasonable to take one concept in this draft, the analysis to varying concepts is our future research.
>
> **Question4:** Very small comment “We leverage u2 − u1 as the change of concept and the label is set as 1 if u2 − u1 is no larger than 0, otherwise the label should be 0.” this should probably be a label for uncertainty, not concept, right?
>
> **Answer:** Thanks for pinning out this issue, the correct sentence should be "We leverage **c2 − c1** as the change of concept and the label is set as 1 if u2 − u1 is no larger than 0, otherwise the label should be 0". We pursue an empirical proof about the connection between activated concepts and uncertainty.

---

### Official Review · Reviewer_4fFT · 2024-11-04

**Soundness:** 2
**Presentation:** 2
**Contribution:** 2
**Rating:** 3
**Confidence:** 3

**Summary:**

This paper examines the impact of multi-round self-reflection prompting without external feedback, termed "Intrinsic Self-Reflection" (ISR), on improving morality-related tasks (e.g., social bias mitigation) for LLMs. The study demonstrates that while ISR significantly enhances performance in both question-answering and text-generation tasks, the improvements diminish after multiple rounds, stabilizing within 10 iterations—an observation termed the "convergence of ISR."

To understand this phenomenon, the authors analyze model uncertainty and calibration errors across ISR rounds. They find that both metrics decrease with more rounds, and the inflection points of calibration error trends coincide with performance stabilization. The authors suggest that ISR reduces model uncertainty and improves calibration, leading to better task outcomes.

The study also probes the model’s hidden states for negative latent concepts such as "immorality" or "toxicity". Results show that positive prompting in the initial round leads to a gradual deactivation (weaker probing performance) of negative concepts over successive rounds.   Injecting immoral instructions in the intermediate rounds only asserts temporary effects, confirming the convergence and irreversibility of ISR.

Finally, through simulation experiments and a theoretical derivation under simplified assumptions, the author bridges the connection between ISR instructions, activated latent concepts, model uncertainty, calibration error, and the converged performance.

**Strengths:**

1. This paper conducts an in-depth study of ISR impact on morality-related tasks for LLMs through the lens of iteration rounds, providing novel insights into deploying ISR to regulate LLM outputs.

2. The organization of this paper is overall clear.

**Weaknesses:**

1. **Confusing Formulation and Non-Rigor Theoretical Investigation**.  The paper contains several ungrounded statements. For instance, in Section 4, the authors attempt to establish causality between model uncertainty, calibration error, and performance to explain ISR convergence. However, the cited previous works and presented empirical evidence only indicate correlation without rigorous analysis or causality testing. Additionally, the assumption of independence between output, instruction, and question in Section 6.2 oversimplifies the language model’s behavior, where outputs are heavily dependent on inputs.  Many notations in Section 6.2 are confusing or redundant  (e.g., redefining conditioning symbol "|"), and Equation 3 misapplies Bayes' rule, leading to problematic derivations. See more details in "Questions".

2. **Inconsistent Experiment Results and Analysis**: There are key discrepancies between experimental outcomes and the authors’ interpretations. For example, VQA performance with more ISR rounds is drops after the second round, challenging the claim that ISR leads to a stable, convergent state. More importantly, in Section 4, as the author uses logit confidence to quantify uncertainty for QA tasks (BBQ here), and "higher logit confidence indicates lower uncertainty" (as explained in Footnote 5 at page 5), Figure 4(a) should be interpreted as QA tasks uncertainty is **increasing** rather than **decreasing** (as in L256), contradicting the whole analysis in Section 4. Also, in Section 5, the "sexual"-dimension of the BBQ dataset exhibits different trends in probing experiments, not consistent with the analysis that negative concepts "converge after several rounds" (hard to read) for "all tasks" (L322).

3. **Confusing Experiment Setup and Missing Experiment Results**: Certain experimental setups are unclear or appear unnecessary For example, the introduction of VQA and visual grounding task (even not clear what this task is without proper citation) seems unnecessary, as the majority of the content of this paper focuses on text-only tasks, and even in the main experiments, there is no related cross-modal discussion. These tasks also seem disconnected from other moral-focused tasks.

    Also, in Section 4, "pick up four social dimensions from the BBQ benchmark (Parrish et al., 2022) for QA tasks" reads confusing – QA tasks, including jailbreak and VQA, have drastically different contexts and it is not clear how BBQ (the "social bias mitigation" task) can be adapted to evaluate "uncertainty" on this task. The best guess here is that other QA tasks are just dropped, but then the jailbreak one is getting back in Section 5 as explained in the incomplete Appendix B.4.

    In Section 5 and Appendix B.4, it seems the jailbreak defense task is adopted in probing experiments, but Figure 5(a) does not apply to jailbreak datasets as these four dimensions are from BBQ and it is not clear how to do this cross-task adaption.

    In Section 6.1, the introduction of the new dataset RealToxicityPrompts is unclear, as there is no related ISR setup explained, and no explanation of how to obtain concepts (c1, c2) and uncertainty (u1, u2) from the trajectories.

**Questions:**

**Confusing Formulation and Non-Rigor Theoretical Investigation**

1. L169 - 175, Eq. 1: The introduction of D as pre-training data is unclear. The derivation of Eq. 1 doesn’t require citing Xie et al. [1]; it follows directly from the law of total probability, which [1] also used.

2. L126 & L169: There is an inconsistency between adopting semantic entropy as an uncertainty measure in L126 and expressing uncertainty as output probabilities in L169. It’s only clarified later that different tasks use different measures.

3. L251: When treating multi-choice QA as classification tasks, and using model log probability for each choice, the counterpart of Semantic Entropy used in the language-generation task here should be direct computation of entropy over re-normalized probabilities for answer choices, rather than using logit confidence. The lack of justification for diverging from entropy-based measures undermines consistency with the language-generation task.

4. L397: "y_t | y_{t-1} denotes that..." – It is not clear what the necessity is to redefine the conditioning notation used in probability.

5. L404: The distinction between M(y_t) and M(y_t | y_{t-1}) is vague.  Also, it is hard to understand how to reach the right-hand side from the left-hand side in this line. One possibility is an unrealistic assumption that improvement in metric M stems from consistently activating more "positive concepts" than "negative" ones in **all** previous rounds preceding round $t$. However, we see that in Figure 5, even sometimes injecting immoral instruction (following the author's reasoning, this means "negative concepts" being activated more), then very likely p(C_n|q_k) > p(C_p | q_k), but later, very quickly, the similarity to toxicity decreases to "moral" line (black line), means that the activation of "positive concepts" are restored. So the improvement at some evaluation rounds does not necessarily mean positive concepts have to be activated more than negative ones all the way.

6. Eq.3: The derivation requires careful review. A detailed proof, reflecting all assumptions, would clarify its validity.

**Experiments**:

1. What is the VQA task you are using? There is a lack of citation.

2. In Figure 5(b), the overlap of the similarity curve with the "moral" curve after introducing intention instructions in rounds 2, 5, and 8 is unclear. The authors state that "immoral instructions drive the activated concept towards toxicity," yet this effect lasts only one round. In contrast, pure immoral/moral instructions seem to have a more prolonged impact. Why the discrepancy?

**Presentation and Writing**:

1. Figure 5(b) – Interve(n)tion Rou(nd), Interven(tion)

2. Appendix B.4 is incomplete and broken.

3. L323: When talking about "injecting immoral instruction/moral instructions", to avoid confusion with later intervention experiments, it should be noted that here the author refers to the black and purple lines.


References:

[1] Xie, Sang Michael, et al. "An Explanation of In-context Learning as Implicit Bayesian Inference." ICLR 2022.

---

> ### Author Response · Authors · 2024-11-22
>
> Dear reviewer, we really appreciate your detailed feedback, motivating us to take more efforts in clarifying some details.
>
> **Weakness1.1**: Confusing Formulation and Non-Rigor Theoretical Investigation. The paper contains several ungrounded statements. For instance, in Section 4, the authors attempt to establish causality between model uncertainty, calibration error, and performance to explain ISR convergence. However, the cited previous works and presented empirical evidence only indicate correlation without rigorous analysis or causality testing.
>
> **Response:** In this draft, we aim to establish connections among self-correction instructions, activated concepts, uncertainty, calibration error, and convergence. This is a complex logical chain and answers why intrinsic self-correction for morality works even though it is superficial[1]. The causal relationship between uncertainty and calibration error is explored by previous studies[2,3,4]. As illustrated in Figure 2, our focus is on bridging the gap between activated concepts and convergence using both empirical and theoretical evidence. We provide empirical support for the relationships between each pair of variables (such as section 6.1 for the variable pair of concept and uncertainty) and offer theoretical evidence to link instructions to convergence. Given the extensive and long chain of the logical reasoning (Figure 2), from the instruction for self-correction to the endpoint of convergence, **we avoid reiterating conclusions from prior studies to ensure the draft remains focused and information-rich**.
>
> [1] Liu, Guangliang, et al. "Intrinsic Self-correction for Enhanced Morality: An Analysis of Internal Mechanisms and the Superficial Hypothesis." Proceedings of the 2024 Conference on Empirical Methods in Natural Language Processing. 2024.
>
> [2] Arendt, Paul D., Daniel W. Apley, and Wei Chen. "Quantification of model uncertainty: Calibration, model discrepancy, and identifiability." (2012): 100908.
>
> [3] Wang, Deng-Bao, Lei Feng, and Min-Ling Zhang. "Rethinking calibration of deep neural networks: Do not be afraid of overconfidence." Advances in Neural Information Processing Systems 34 (2021): 11809-11820.
>
> [4] Ao, Shuang, Stefan Rueger, and Advaith Siddharthan. "Two sides of miscalibration: identifying over and under-confidence prediction for network calibration." Uncertainty in Artificial Intelligence. PMLR, 2023.
>
> **Weakness1.2**: Additionally, the assumption of independence between output, instruction, and question in Section 6.2 oversimplifies the language model’s behavior, where outputs are heavily dependent on inputs. Many notations in Section 6.2 are confusing or redundant (e.g., redefining conditioning symbol "|"), and Equation 3 misapplies Bayes' rule, leading to problematic derivations. See more details in "Questions".
>
> **Response:** **We did not assume independence between output and input** at each self-correction round. But we only have independence assumption among question, instruction and previous responses.  Since we frame self-correction as a format of QA, therefore, at each self-correction round, the input includes question, self-correction instruction and dialog history wherein the previous responses exist.
>
> We have the independence assumption because
>
> (1) currently the research community still does not have a solid theoretical tool to interpret LLMs which remain a black-box for us. In-context learning is an example, some studies consider it as a mesa optimization but others believe it implements a meta-learning process. [1] proposes a theoretical analysis to the self-correction by assuming self-correction process implements a Bradley-Terry model through gradient descent. The authors have a strong and only theory-driven assumption about the equivalence between LLMs' (forward) inference and gradient descent. In our case, the assumption of independence is the most optimal and effective one we can make right now.
>
> (2) we do not pursue to have an accurate mathematical modeling for how self-correction works, which requires more efforts. Instead, based on our empirical observations, we aim at having a theoretical modeling to link the activated concept (self-correction instruction) and the convergence of intrinsic self-correction. Therefore, we can ensure the relationship between activated concept and the convergence of self-correction is validated both empirically and theoretically.
>
> [1] Wang, Yifei, et al. "A Theoretical Understanding of Self-Correction through In-context Alignment." arXiv preprint arXiv:2405.18634 (2024).

---

> ### Author Response · Authors · 2024-11-22
>
> **Weakness2.** Inconsistent Experiment Results and Analysis: There are key discrepancies between experimental outcomes and the authors’ interpretations.
>
> **Response:** Thanks for your question. After carefully reviewing our statements, we did not find inconsistencies between experiment results and analysis. Instead, there seems to have an mis-communication due to some terminology without well explanations. We will clarify these terms and address them appropriately in the revised version.
>
> **Weakness2.1**: VQA performance with more ISR rounds is drops after the second round, challenging the claim that ISR leads to a stable, convergent state.
>
>
> **Response:** The stable, convergent state should meet two requirements: (1) the model performance no longer changes drastically and (2) the convergent performance demonstrates a performance gain over the baseline, where no self-correction instructions are applied.
>
> It is important to note that convergence does not necessarily result in the best performance. For instance, in the case of a vanilla MLP, different training configurations can yield varying different performance, yet they can achieve stable convergence without encountering training failures, such as gradient vanishing or explosion.
>
>
>
> **Weakness2.2**: In Section 4, as the author uses logit confidence to quantify uncertainty for QA tasks (BBQ here), and "higher logit confidence indicates lower uncertainty" (as explained in Footnote 5 at page 5), Figure 4(a) should be interpreted as QA tasks uncertainty is increasing rather than decreasing (as in L256), contradicting the whole analysis in Section 4.
>
>
> **Response:** Thanks for your careful review and identify the potential confusion. The uncertainty metric we calculate for the QA task is defined as (1 - ECE score) , ensuring consistency in the observed trends. The QA task demonstrates high confidence (converge to 1) with consistent performance. We have add corresponding explanation in our paper revision.
>
> **Weakness2.3** in Section 5, the "sexual"-dimension of the BBQ dataset exhibits different trends in probing experiments, not consistent with the analysis that negative concepts "converge after several rounds" (hard to read) for "all tasks" (L322).
>
> **Response2.3**: Similar to the response in 2.1, convergence does not necessarily imply that the entire process is exactly the same. However, all curves exhibit stable behavior without significant fluctuations after the third round.
>
> **Weakness3.1**: the introduction of VQA and visual grounding task (even not clear what this task is without proper citation) seems unnecessary, as the majority of the content of this paper focuses on text-only tasks, and even in the main experiments, there is no related cross-modal discussion. These tasks also seem disconnected from other moral-focused tasks.
>
> **Response:** Thanks for your suggestions. We have add more citations on VQA and Visual grounding in the revision. Detailed experimental setup can also be found at the first paragraph of Section 3.
>
> The multi-modal tasks is a side evidence to demonstration the convergence of LLMs. Notably, our whole paper focuses on LLMs as we almost do not mention any multi-modality and VLM. We would like to leave the further discuss on the cross-modal discussion as a promising future work.
>
> **Weakness3.2**: in Section 4, "pick up four social dimensions from the BBQ benchmark (Parrish et al., 2022) for QA tasks" reads confusing – QA tasks, including jailbreak and VQA, have drastically different contexts and it is not clear how BBQ (the "social bias mitigation" task) can be adapted to evaluate "uncertainty" on this task. The best guess here is that other QA tasks are just dropped, but then the jailbreak one is getting back in Section 5 as explained in the incomplete Appendix B.4.
>
> **Response:** Regarding multi-choice QA tasks, we consider LLMs’ predictions as a classification problem, therefore leveraging the ECE error. How to measure the uncertainty of the QA task can be found in line 249-251.  The additional experiment on Jailbreak can be found in appendix B.3 with consistent observations.
>
> **Weakness3.3**: In Section 5 and Appendix B.4, it seems the jailbreak defense task is adopted in probing experiments, but Figure 5(a) does not apply to jailbreak datasets as these four dimensions are from BBQ and it is not clear how to do this cross-task adaption.
>
> **Response:** Thanks for your suggestion to help our paper more comprehensive. We have add the probing experiment Jailbreak in Appendix B.3. The same observations were noted, further supporting the indication of cross-task adaptation.
> As for the selection of four dimensions from BBQs, this choice was made because the probe values across different tasks have entirely different scales, making it challenging to visualize them within a single figure.

---

> > ### Author Response · Authors · 2024-11-22
> >
> > **Weakness3.4**: In Section 6.1, the introduction of the new dataset RealToxicityPrompts is unclear, as there is no related ISR setup explained, and no explanation of how to obtain concepts (c1, c2) and uncertainty (u1, u2) from the trajectories.
> >
> > **Answer:** Sorry for the inconvenience. We have revised it. The RealToxicityPrompts is the text detoxification task, a.k.a. the language generation task, we presents in Figure 4 and Figure 5. For measuring the concept, we leverage the linear probing vector acquired from a toxicity classification task and we took the semantic uncertainty to estimate the model uncertainty. The prompting format is available in Appendix B.5.
> >
> > **Question1:** theory. Questions:
> > Confusing Formulation and Non-Rigor Theoretical Investigation
> >
> > L169 - 175, Eq. 1: The introduction of D as pre-training data is unclear. The derivation of Eq. 1 doesn’t require citing Xie et al. [1]; it follows directly from the law of total probability, which [1] also used.
> >
> > **Response:** The pre-training data D here is only for indicting the source of learned concept. But, we do not know ditribution or any other characteristics of the pre-training data. The reason that we cite Xie et al. is the authors establish a foundation for analyzing LLMs by referencing the semantic space of concepts, simplifying the process of in-context learning. Alternatively, we might need to begin with an analysis of the token space and validate the accuracy of the concepts before engaging in more in-depth discussions.
> >
> > **Question2:** L126 & L169: There is an inconsistency between adopting semantic entropy as an uncertainty measure in L126 and expressing uncertainty as output probabilities in L169. It’s only clarified later that different tasks use different measures.
> >
> > **Response:** Thanks for this great suggestion, we are sure to make it clear in our future version.
> >
> > **Question3:** L251: When treating multi-choice QA as classification tasks, and using model log probability for each choice, the counterpart of Semantic Entropy used in the language-generation task here should be direct computation of entropy over re-normalized probabilities for answer choices, rather than using logit confidence. The lack of justification for diverging from entropy-based measures undermines consistency with the language-generation task.
> >
> > **Response:** The semantic uncertainty is only used for language generation tasks but not for multi-choice QA tasks. For the multi-choice QA task, we take the prediction logits approximated by log likelihood as the uncertainty.
> >
> > **Question4:** L397: "y_t | y_{t-1} denotes that..." – It is not clear what the necessity is to redefine the conditioning notation used in probability.
> >
> > **Response:** As we highlight in the line of 398, we tend to emphasize that the qualify of y_t depends on y_{t-1}, and multi-round self-correction is a natural choice and we focus on convergence for this particular reason.
> >
> > **Question5:** L404: The distinction between M(y_t) and M(y_t | y_{t-1}) is vague. Also, it is hard to understand how to reach the right-hand side from the left-hand side in this line. One possibility is an unrealistic assumption that improvement in metric M stems from consistently activating more "positive concepts" than "negative" ones in all previous rounds preceding round
> > . However, we see that in Figure 5, even sometimes injecting immoral instruction (following the author's reasoning, this means "negative concepts" being activated more), then very likely p(C_n|q_k) > p(C_p | q_k), but later, very quickly, the similarity to toxicity decreases to "moral" line (black line), means that the activation of "positive concepts" are restored. So the improvement at some evaluation rounds does not necessarily mean positive concepts have to be activated more than negative ones all the way.
> >
> > **Response:** We take the term M(y_t | y_{t-1}) to highlight that the quality of y_t depends on y_{t-1}, in a multi-round QA scenario.
> >
> > The red line in Figure 5.b is utilized to hilight the **irreversibility property of activated concepts**. We only show how the activated concepts would be intervened if immoral instructions are introduced in each of particular rounds of **normal self-correction with moral instructions**. However, the purple line (immmoral) clearly indicate the activated toxicity within hidden states since the 2nd round.
> >
> > **Question6:** Eq.3: The derivation requires careful review. A detailed proof, reflecting all assumptions, would clarify its validity.
> >
> > **Answer:** Thanks for this great suggestion. Yes, we will improve the derivation by adding that x, i and y are indpendent conditional on positive concept C_p: p(x,i,y|C_p) = p(x|C_p)p(i|C_p)p(y|C_p).

---

> ### Author Response · Authors · 2024-11-22
>
> **Experiments:**
>
> **Question 1:** What is the VQA task you are using? There is a lack of citation.
>
> **Response:** Details can be found in [1]. We have added corresponding citation in our paper.
>
> [1] Shengbang Tong, Zhuang Liu, Yuexiang Zhai, Yi Ma, Yann LeCun, and Saining Xie. Eyes wide shut? exploring the visual shortcomings of multimodal llms, 2024.
>
> **Question 2:** In Figure 5(b), the overlap of the similarity curve with the "moral" curve after introducing intention instructions in rounds 2, 5, and 8 is unclear. The authors state that "immoral instructions drive the activated concept towards toxicity," yet this effect lasts only one round. In contrast, pure immoral/moral instructions seem to have a more prolonged impact. Why the discrepancy?
>
> **Response:** Sorry for the confusion about the overlapping part. In this line, we only have three valid point, which is rounds 2, 5, 8. We are not add intevention on 2, 5, 8 round simultaneously in one experiment. We are implemening three experiments with intevention on 2, 5, 8 round individually. For the overlapping part, it is just the same as the original experiments without intenvention. We will add more explanations in the revision.

---

> > ### Author Response · Authors · 2024-11-26
> >
> > Dear Reviewer, as the discussion period is nearing its end, we would be happy to address any remaining questions you may have.

---

> ### Comment · Reviewer_4fFT · 2024-11-27
> **Replies to the responses (1/3)**
>
> Thanks to the authors for the detailed responses. Here are my replies:
>
> - Re: Weakness 1.1:
>
> It seems there is a misunderstanding. I am not asking to reiterate prior conclusions in my review. My concern is that most prior studies (including the ones the authors cited) do not establish a "bi-directional causal relation" (L262 in the revision). They typically highlight **correlation** between `uncertainty` and `confidence` under specific definitions, which do not align with the definitions in this paper.
>
> Let's take one most recent citation [1] (the authors cited as "[3]") for example. [1] states in Section 3.2 that traditional ECE "fails to show whether the calibration error is caused by under- or over-confident predictions," (and more discussions of ECE limitation in Section 2.2). This statement already breaks the claimed "bi-directional causality." Additionally, their "confidence" is defined as bin-wise average probability (Section 3.1), not the instance-wise logit confidence defined in this paper. Their experiments focus on vision models and small-scale language models, raising doubts about their transferability to large models, which are known to face severe miscalibration issues (e.g., GPT-4 [2]).
>
> **While proving this causality is not the focus of the paper, such unsupported claims ("bi-directional causality") suggest a lack of rigor and respect for prior work.**
>
> - Re: Weakness 1.2
>
> The authors claim they do not assume independence between input and output. However, L399 states:
> > We have the independence assumption over question x, instruction i and output y, e.g., p(x, i, y) = p(x)p(i)p(y)
>
> and L403 adds:
> > Another assumption is x, i, y are independent conditional on C_p, i.e., p(x, y, i|C_p) = p(x|C_p)p(y|C_p)p(i|C_p)
>
> The second one might make some sense, as it assumes a latent variable that disentangles input, output, and instruction. But the first one is contradictory to what the authors emphasized. However, the first assumption contradicts the authors' claim of avoiding independence assumptions. Furthermore, no notation accounts for "previous responses" as suggested in the rebuttal, making the statement "we only have independence among question, instruction, and previous responses" unclear (and there are two strong assumptions here, not only one).
>
> Also the statement in the rebuttal "we only have independence assumption among question, instruction and previous responses" is confusing -- the generation of "previous responses" should be conditioned on the question and the instruction, how can these three be independent?
>
> Regarding the authors' justification for the independence assumption:
>
> > (1) currently the research community still does not have a solid theoretical tool to interpret LLMs which remain a black-box for us. In-context learning is an example, .... The authors have a strong and only theory-driven assumption about the equivalence between LLMs' (forward) inference and gradient descent. In our case, the assumption of independence is the most optimal and effective one we can make right now.
>
> The authors argue that since LLMs are black-box models, strong assumptions are permissible, citing analogies like ICL-as-GD (in-context learning as gradient descent). I am afraid this is another baseless statement. We can always make assumptions for the ease of theoretical analysis, but this comes with the cost of not reflecting the truth and undermining the applicability of the analysis. Unfortunately, for this paper, the assumption made is not reasonable and can lead to significant errors.
>
> > (2) ...Instead, based on our empirical observations, we aim at having a theoretical modeling to link the activated concept (self-correction instruction) and the convergence of intrinsic self-correction. Therefore, we can ensure the relationship between activated concept and the convergence of self-correction is validated both empirically and theoretically.
>
> The authors claim their work bridges empirical observations with theoretical modeling. However, this seems closer to "simulation" or "interpretation" rather than rigorous theoretical derivation.
>
>
> **References**
>
> [1] Ao, Shuang, Stefan Rueger, and Advaith Siddharthan. "Two sides of miscalibration: identifying over and under-confidence prediction for network calibration." Uncertainty in Artificial Intelligence. PMLR, 2023.
>
> [2] Achiam, Josh, et al. "GPT-4 technical report." arXiv preprint arXiv:2303.08774 (2023).

---

> ### Comment · Reviewer_4fFT · 2024-11-27
> **Replies to the responses (2/3)**
>
> 1. Re: Weakness 2: Thank you for the extra effort in revising the terminology. While progress has been made, the terminology throughout the paper still requires more careful examination.
>
> 2. Re: Weakness 2.1: While I agree that "stability" does not imply better performance, my original concern was that VQA performance **continues to drop**, and it is unclear whether the trend is truly "converging." The analogy to gradient descent for MLP training is intriguing, but optimization-via-prompting in discrete space differs fundamentally from optimization in continuous space. It is unclear what the discrete optimization counterpart for "gradient vanishing or explosion" would be and we should be caution about this analogy.
>
> 3. Re: Weakness 2.2: Thanks for the clarification. I noted that the author added "Uncertainty task for QA tasks corresponds to 1 - ECE score" in Figure 4. However this raises another severe problem -- if Figure 4(a) uses 1-ECE for uncertainty, the trend in 4(a) should inversely mirror the ECE trend in Figure 4(b).  This inconsistency needs further explanation. Also, the choice to report 1-ECE as uncertainty measure is not common -- the authors should add citation and corresponding justification.
>
> 4. Re: Weakness 2.3: I am not sure whether the trend showing "decreasing-then-increasing" can be regarded as "converging" (L323). Also, my original comment is mostly on the wording of "all tasks" -- if there are some nuanced differences, perhaps it should be tempered a bit.
>
> 5. Re: Weakness 3.1 (The necessity of introducing VQA tasks):
> > The multi-modal tasks is a side evidence to demonstration the convergence of LLMs. Notably, our whole paper focuses on LLMs as we almost do not mention any multi-modality and VLM. We would like to leave the further discuss on the cross-modal discussion as a promising future work.
>
>     I am not sure whether the authors' response justifies the necessity of introducing VQA tasks. As the author admits, the whole paper has a significant focus on LLMs and there is not much analysis on the vision side. The experiments on vision only serve as "side evidence". Removing the whole vision part to the Appendix does not seem to interfere with the main results and analysis in the paper.
>
> 6. Re: Weakness 3.2 & 3.3 (Uncertainty Evaluation on QA tasks for jailbreak and VQA): I reviewed Appendix B.3 in the revision (Second Revision to date) and want to issue a **WARNING** to AC/SAC and other readers: **The authors directly copied my comments from Weakness #3 word-for-word into their paper without my permission**, presenting them as their own: "The additional experiment on Jailbreak can be found in Appendix B.3 with consistent observations." This raises serious concerns about academic integrity.
>
> 7. Re: Weakness 3.4: Thanks for the additional explanation. However, Appendix B.5 is not the prompting format the authors referred to, and it is not clear which of these three paragraphs points to RealToxicityPrompts.

---

> ### Comment · Reviewer_4fFT · 2024-11-27
> **Replies to responses (3/3), Summary**
>
> For other questions regarding theory, writing, and presentation, I believe the authors are making progress, but not all promised improvements are reflected in the current revision. More thorough checks on terminology and theoretical derivations are still required.
>
> Overall, after reviewing the responses, I feel even less confident about the acceptance of the paper and will not raise my scores. While I appreciate the authors' efforts in providing detailed responses and improving the writing, I noticed several severe new issues in my earlier responses:
>  (1) non-factual reference to previous works, 2) unrealistic independence assumption, 3) suspicious arguments, measurements and figures, 4) potential overclaim, 5) direct copy of my review comments and unfaithful report of revision in Appendix B.3 and B.5.
>
> Despite these concerns, I will not further lower my scores and will maintain them as-is.

---

> ### Author Response · Authors · 2024-11-28
> **Our new response to Replies to the responses (1/3)**
>
> **Weakness 1.1**
>
> **Response:** With the term **bi-directional** causality, we would like to highlight that: the calibration error reduces, but we can not determine the model is less confident or more confident since the decrease of calibration error can happen on both conditions that the model is less over-confident and less under-confident. Although there is a causal relationship between calibration error and confidence, the status of these two variables cannot be determined solely by examining one in relation to the other. We are sure to add more details to make this term more clear.
>
> Please note we calculate ECE error in a bin way, the point is, in previous literatures, people tend to use confidence and uncertainty interchangeably. In this draft, we carefully take the term prediciton logit confidence if we need to hihlight the logit, in order to avoid misusing confidence and uncertainty.
>
>
> **Weakness 1.2 and the independence assumption**
>
> **Response:** Please note we have a multi-round QA interaction scenario. As we highlight in our previous rebuttal, the response y at round t (y_t), is generated according to previous dialog history from round 1 to round t-1, and the instruction for round t. This is the reason that we can have independence assumption over input question x and response y, since x is only used in the 1st round. The equation 3 clearly indicates this. In the draft, we carefully take the term **formulation** instead of claiming we achieved a concrete mathematical proof. Our result is in line with our empirical evidence.

---

> ### Author Response · Authors · 2024-11-28
> **Clarify for the error**
>
> Dear reviewer,
>
> Please accept our sincere apologies for the unintentional mistake. We copied your comments into our draft as a reminder to include a figure for your question but mistakenly forgot to remove it. Since your comment was not intended as an answer to your question and we should have a figure to answer the question, we would never have included it intentionally. We deeply regret the confusion this error may have caused.
>
> Even though you said you would not change your score, we are still happy to address your questions to make this draft better.

---

### Meta-Review · Area_Chair_5a2q · 2024-12-28

**Metareview:**

This paper investigates the ISR impact on LLMs for morality-related tasks. The reviewers note that the metrics had some uncertainty, the experimental scope is limited, and the paper generally would benefit form further work.

**Additional Comments On Reviewer Discussion:**

I do not believe there is an instance of misconduct in this discussion, and my decision is independent of that conversation.

---

### Decision · Program_Chairs · 2025-01-22

Reject